# A Phase-I pharmacokinetic, safety and food-effect study on flubentylosin, a novel analog of Tylosin-A having potent anti-*Wolbachia* and antifilarial activity

**Negar Alami** [1,2]*, **David C. Carter** [1,3], **Nisha V. Kwatra** [1,4], **Weihan Zhao** [1], **Linda Snodgrass** [1], **Ariel R. Porcalla** [1], **Cheri E. Klein** [1], **Daniel E. Cohen** [1], **Loretta Gallenberg** [1,3], **Melina Neenan** [1], **Robert A. Carr** [1,3], **Kennan C. Marsh** [1], **Dale J. Kempf** [1,3]

**1** AbbVie, North Chicago, Illinois, United States of America, **2** Pfizer, Chicago, Illinois, United States of America, **3** Retirees of AbbVie, Chicago, Illinois, United States of America, **4** Food and Drug Administration, Silver Spring, Maryland, United States of America

* nalami03@gmail.com

**Data Availability Statement:** All relevant data are within the manuscript and its Supporting Information files.

## Abstract

### Background

The parasitic filariae responsible for onchocerciasis and lymphatic filariasis are host to an endosymbiotic bacterium, *Wolbachia*, which is essential to the fertility and development of the parasites. We performed a Phase-I pharmacokinetic, safety and food-effect study on single and multiple ascending doses of flubentylosin (ABBV-4083), a macrolide antibacterial with activity against *Wolbachia*, intended to sterilize and eliminate the parasites.

### Methods

Seventy-eight healthy adults were exposed to flubentylosin; 36 were exposed to single ascending 40, 100, 200, 400 or 1000 mg doses; 12 received 1000 mg in the food-effect part; and 30 received multiple ascending daily doses of 100 mg for 7 days, 200 mg for 7 or 14 days, or 400 mg for 7 or 14 days. Twenty-two subjects received placebo.

### Results

Maximum concentrations ($C_{max}$) of flubentylosin were reached after 1–2 hours, with a half-life < 4 hours at doses ≤ 400 mg. $C_{max}$ and AUC increased in a more than dose-proportional manner, with similar exposure after multiple dose administration. The most frequently reported adverse events were nausea (8/78, 10%) and headache (6/78, 8%).

Two subjects given a single dose of flubentylosin 1000 mg in the food-effect part experienced reversible asymptomatic ALT and AST elevations at Grade 2 or Grade 4, with no elevation in bilirubin, deemed related to study drug. The effect of food on exposure parameters was minimal. No treatment-related serious adverse events were reported.

**Funding:** All authors are or were employees of AbbVie and may own AbbVie stock. AbbVie sponsored and funded the study; contributed to the design; participated in the collection, analysis, and interpretation of data, and in writing, reviewing, and approval of the final publication.

**Competing interests:** I have read the journal's policy and the authors of this manuscript have the following competing interests: All authors are or were employees of AbbVie during the work of this study and may own AbbVie stock. AbbVie sponsored and funded the study; contributed to the design; participated in the collection, analysis, and interpretation of data, and in writing, reviewing, and approval of the final publication. DJK, LG and RAC are retirees of AbbVie. NVK is currently an employee of the Food and Drug Administration. NNA is currently an employee of Pfizer.

## Discussion

Flubentylosin 400 mg for 14 days was the maximum tolerated dose in this first-in-human, Phase-I study in healthy adults. Based on preclinical pharmacokinetic/pharmacodynamic modeling, flubentylosin 400 mg once daily for 7 or 14 days is expected to be an effective dose. A Phase-II, proof-of-concept study with flubentylosin using these regimens is currently ongoing in patients with onchocerciasis in Africa.

## Author summary

Onchocerciasis and lymphatic filariasis are neglected tropical diseases caused by parasitic filarial nematodes. Current efforts to eliminate these diseases are hindered by a lack of drugs that permanently sterilize and/or kill the adult worms. Antibacterials, including doxycycline, have been shown to deplete *Wolbachia*, an endosymbiotic bacterium essential to the fertility and development of the adult worms, leading to their permanent sterilization and death. However, doxycycline is contraindicated in women of child-bearing age, breastfeeding women, and children, and must be given for 4–6 weeks to be effective. There is a need for agents with fewer contraindications and a shorter treatment regimen. Flubentylosin is an antibacterial with demonstrated anti-*Wolbachia* activity in several animal models of filarial disease. The present first-in-human Phase-I clinical study investigated the safety and pharmacokinetics of flubentylosin in healthy male and female subjects after single and multiple ascending doses, as well as the effect of food on the pharmacokinetics of flubentylosin, in order to identify appropriate regimens for Phase-II studies. Treatment regimens with 400 mg flubentylosin for 7 or 14 days were selected for further investigation.

## Introduction

The filarial diseases onchocerciasis ("river blindness") and lymphatic filariasis (LF) represent major health problems in affected countries, responsible for respectively 1.34 and 1.36 million disability-adjusted life years lost in 2017 [1].Both diseases are caused by parasitic worms transmitted by the bite of blood-feeding insects (the vector), i.e. blackflies of the genus *Simulium* in onchocerciasis and Anopheles mosquitoes for LF. The parasitic filariae in question, *Onchocerca volvulus* in the case of onchocerciasis, and *Wuchereria bancrofti*, as well as *Brugia* spp., in the case of LF, are transmitted in larval form to humans during a blood meal of the vector. In both diseases, the larvae mature into reproductively competent adults within the human host and produce thousands of progeny (microfilariae). Onchocerciasis results from the death of these microfilariae, which causes an inflammatory response through the release of filarial and bacterial antigens, leading to skin rash, lesions, itching and skin depigmentation, as well as local ocular complications, often leading to blindness. In an *in vivo* ocular challenge model, it was shown that prior depletion of wolbachial antigen from the filarial antigen led to a much reduced inflammatory reaction in the eyes, suggesting an additional role for *Wolbachia* in the disease symptoms [2]. In LF, adult worms (macrofilariae) migrate to the lymphatic vessels, and the resulting impairment of lymphatic drainage leads to lymphoedema, elephantiasis and hydrocoele, the most common chronic disabling complications. The microfilariae circulate in

the blood at night in LF and during the day in onchocerciasis, where they are taken up by the vector and passed on to other humans [3].

Control and treatment of the two diseases currently rely on mass drug administration (MDA), with ivermectin for onchocerciasis in sub-Saharan Africa, ivermectin in association with albendazole for LF in sub-Saharan Africa, and diethylcarbamazine in association with albendazole in other parts of the world [4,5]. These products are, however, mainly microfilaricidal. They temporarily impair embryo production but do not kill the adult worms and therefore need to be administered on an annual or semi-annual basis over a 10- to 15-year period to achieve long-term control. In addition, in areas co-endemic for infection with the filarial parasite *Loa loa*, the risk of serious adverse events related to the death of *Loa* microfilariae limits the use of ivermectin [6]. Recent data on triple-drug therapy with ivermectin, albendazole and diethylcarbamazine have shown considerable promise in LF, however transposition to onchocerciasis may prove challenging [7].

Moreover, MDA entails considerable logistic efforts and financial commitments, and there are ethical considerations regarding treating entire populations, including uninfected individuals. Cost effectiveness may also be an issue since success requires sustained coordination of supply logistics to ensure that treatment reaches all target recipients. Because of these challenges, interruption of transmission and disease elimination has proven more difficult than initially expected. In addition, MDA given for long periods carries the risk of resistance, described for ivermectin in animal health use, together with suboptimal responses in humans [8]. The World Health Organization has set an ambitious goal of eliminating onchocerciasis in 80% of African countries by 2025 [9], and the United Nations Sustainable Development Goals include a 90% reduction in the number of persons requiring intervention by 2030 [10]. A number of studies and simulations indicate that [11–13], with current strategies, these goals are not feasible. Alternative approaches are clearly needed.

The parasitic filariae responsible for onchocerciasis and for LF are host to an endosymbiotic bacterium, *Wolbachia*, that is essential to the fertility and development of the parasites. Depletion of *Wolbachia* using antibacterial therapy, such as the tetracycline antibacterials doxycycline and minocycline, has demonstrated efficacy in treating these diseases in clinical studies [5,14–15]. Use of these agents is, however, limited and not ideal for broad implementation due to contraindications in women of child-bearing age, in breastfeeding women and in children. In addition, their relatively long treatment duration, i.e. 5–6 weeks in onchocerciasis and 4 weeks in LF, can result in sub-optimal treatment adherence.

The anti-*Wolbachia* (A·WOL) consortium, made up of academic and industrial partners, was formed with the long-term goal of discovering and developing novel anti-*Wolbachia* agents with superior profiles. Screening of focused libraries, including screening of a library of anti-infectives provided by Abbott (now AbbVie), led to the discovery that the veterinary antibiotic, tylosin, had potent anti-*Wolbachia* activity. This prompted a medicinal chemistry programme at AbbVie that resulted in the development of a new next-generation antifilarial agent.

Flubentylosin (ABBV-4083), a semisynthetic analog of the macrolide antibiotic tylosin A, demonstrated strong potential as an anti-wolbachial agent in preclinical studies, in terms of its activity and safety, and a possible reduction in treatment duration compared to doxycycline [16,17]. As a result of its anti-wolbachial activity, flubentylosin indirectly targets the macrofilariae, causing a loss of fertility in adult female worms that leads to reduced levels of microfilariae. Indeed, in mouse and gerbil infection models of onchocerciasis and LF, flubentylosin achieved > 90% depletion of *Wolbachia* and clearance of microfilarial infection comparable or superior to the reference tetracycline antibacterials [16,17]. Studies on the *in vivo* kinetics of *Wolbachia* depletion by flubentylosin showed that its activity is both dose- and treatment

duration-dependent, providing the basis for investigation of 7- and 14-day treatment regimens in the clinical setting [7]. In preparation for planned efficacy studies in onchocerciasis-endemic settings, we describe here the results of a first-in-human, Phase-I, single- and multiple-ascending-dose and food effect study to investigate the safety and pharmacokinetics of flubentylosin in healthy subjects, which enabled us to identify a dose and dosing regimen for use in subsequent studies.

## Methods

### Ethics statement

The study was conducted in accordance with International Council for Harmonisation of Technical Requirements for Pharmaceuticals for Human Use (ICH) guidelines on good clinical practice (GCP), applicable regulations governing clinical study conduct and the ethical principles that have their origin in the Declaration of Helsinki. As per GCP, an institutional review board, Quorum (approval number: 32868/1), ensured the ethical, scientific, and medical appropriateness of the study before it was initiated and approved all relevant documentation.

Written informed consent was obtained from all subjects before undertaking any study-related procedures.

The study was conducted under the direction of AbbVie, who was responsible for study monitoring and flubentylosin concentration assays (blood and urine). Clinical laboratory tests and safety data management were performed by contract research organizations (Chicago Clinical Laboratories Ltd, Northbrook IL USA and Syneos Health, Morrisville NC USA).

### Study design

This was a three-part, single-center, first-in-human Phase-I study performed in healthy subjects. The design features for the three parts of the study are summarized in Table 1, including single ascending dose (Part 1), food effect (Part 2) and multiple ascending dose (Part 3). The initial doses of flubentylosin were selected based on preclinical toxicology studies. The dog was the most sensitive species with a no-observed-adverse-effect level of 15 mg/kg/day which, after applying a conversion factor, translated to 450 mg in a 60-kg human. The ascending dose cohorts in Parts 1 and 3 were enrolled following review of the safety and pharmacokinetic data from the previous cohorts. Part 2 included two periods and sequences (fed-fasting or fasting-fed).

### Study Drug

Flubentylosin and matching placebo were supplied as 10 and 100 mg capsules.

### Subjects

Adult male or female subjects in good general health were selected for enrollment in the study based on selection criteria and randomized to flubentylosin or placebo in Parts 1 and 3, and to fed or fasting conditions in Period 1 of Part 2.

Key eligibility criteria included age between 18 and 55 years inclusive at the time of enrollment, body mass index $\geq 18.0$ to $\leq 29.9$ kg/m$^2$, postmenopausal or surgically sterile or using effective double contraception during the study and for 90 days after the last intake of study drug, normal liver enzyme levels at screening and enrollment and negative results for hepatitis A, B and C, as well as human immunodeficiency virus.

**Table 1. Overview of the Three Parts of the Phase-1 First-in-human Study on Flubentylosin.**

| | Part 1—Single Ascending Dose Study | Part 2—Food Effect Study | Part 3—Multiple Ascending Dose Study |
|---|---|---|---|
| **Design features** | Randomized, double-blind, placebo-controlled, single-ascending-dose study | Randomized, open-label, two-sequence, two-period, single-dose crossover study | Randomized, double-blind, placebo-controlled, multiple-ascending-dose study |
| **Study groups** | 6 groups of 8 subjects each; 6 on flubentylosin, 2 on placebo in each group | 1 group of 12 subjects; randomized to Sequence 1 (Period 1 fasting → Period 2 fed) or Sequence 2 (Period 1 fed → Period 2 fasting) | 5 groups of 8 subjects each; 6 on flubentylosin, 2 on placebo in each group |
| **Study population** | Healthy adult subjects | Healthy adult subjects | Healthy adult subjects |
| **Objectives** | Assess PK and safety of single ascending doses of flubentylosin | Assess the food effect on the PK and safety of flubentylosin | Assess PK and safety of multiple ascending doses of flubentylosin |
| **Doses studied** | Single intake<br>Group 1: 40 mg (4 ´ 10 mg capsule)<br>Group 2: 100 mg (1 ´ 100 mg capsule)<br>Group 3: 200 mg (2 ´ 100 mg capsule)<br>Group 4: 400 mg (4 ´ 100 mg capsule)<br>Group 5: 1000 mg (10 ´ 100 mg capsule)*<br>Group 6: 400 mg (4 ´ 100 mg capsule)** | Group 7: 1000 mg (10 ´ 100 mg capsule)* | Group 8: 100 mg QD (1 ´ 100 mg capsule) for 7 days<br>Group 9: 200 mg QD (2 ´ 100 mg capsule) for 7 days<br>Group 10: 200 mg QD (2 ´ 100 mg capsule) for 14 days<br>Group 11: 400 mg QD (4 ´ 100 mg capsule) for 7 days<br>Group 12: 400 mg QD (4 ´ 100 mg capsule) for 14 days |
| **PK sample collection** | Pre-dose, and at 0.5, 1, 1.5, 2, 3, 4, 5, 6, 8, 12, 15, 24, 30, 36, 48, 60 and 72 hours after dosing | Pre-dose, and at 0.5, 1, 1.5, 2, 3, 4, 5, 6, 8, 12, 15, 24, 30, 36, 48, 60 and 72 hours after dosing | All groups: Day 1 pre-dose, and at 0.5, 1, 1.5, 2, 3, 4, 5, 6, 8, 12, 15, and 24 hours after dosing<br>Groups 8, 9 and 11: pre-dose on Days 3, 4, 5 and 6; Day 7 pre-dose, and at 0.5, 1, 1.5, 2, 3, 4, 5, 6, 8, 12, 15, 24, 30, and 36 hours after dosing;<br>Groups 10 and 12: pre-dose on Days 7, 11, 12, 13; Day 14 pre-dose, and at 0.5, 1, 1.5, 2, 3, 4, 5, 6, 8, 12, 15, 24, 30, and 36 hours after dosing |

QD; once daily; PK: pharmacokinetics

* 800 mg (8 ´ 100 mg capsule) planned in protocol, adjusted to 1000 mg based on pharmacokinetic and safety results of previous groups in Part 1

** 1200 mg (12 ´ 100 mg capsule) planned in protocol, changed to 400 mg with confinement extended from 5 to 12 days and additional laboratory test

## Pharmacokinetic analyses

Flubentylosin concentrations in blood and urine were determined using validated liquid chromatography methods with tandem mass spectrometric detection. The lower limit of quantitation for flubentylosin was 1.00 ng/mL in blood and 5.00 ng/mL in urine. Samples quantified below the lowest standard were reported as zero.

The maximum observed blood concentration ($C_{max}$) and the time to $C_{max}$ ($T_{max}$) were determined directly from the blood concentration-time data. The value of the apparent terminal phase elimination rate constant (beta, β) was obtained from the slope of the least squares linear regression of the logarithms of the blood concentration-time data from the terminal log-linear phase of the profile. The area under the blood concentration-time curve (AUC) from time 0 to the time of the last measurable concentration ($AUC_t$) was calculated by the linear trapezoidal rule. The AUC was extrapolated to infinity by dividing the last measurable blood concentration by β. For Part 3, the observed blood concentration at 24 hours after dosing ($C_{24}$) was presented. AUC during a dosing interval ($AUC_{24}$) was determined using noncompartmental methods. Dose-normalized $C_{max}$ and dose-normalized AUC were calculated in Parts 1 and 3.

## Safety assessments

Safety was assessed by monitoring adverse events (AEs), changes in vital signs, physical examinations, electrocardiogram, and laboratory test assessments.

## Statistics

Statistical analyses were performed using SAS version 9.2 (Cary, NC, USA) or higher under the UNIX operating system. For each study part, descriptive statistics were provided for demographic variables by flubentylosin dose group. Safety data were tabulated by number of subjects and percentage. PK parameters were derived from blood concentration versus time data using a noncompartmental analysis in SAS version 9.2 (Cary, North Carolina, USA). Concentration versus time data in blood and PK parameters were listed and summarized by regimen, using descriptive statistics. Planned sampling times were used in summaries and plots. If the actual sampling was within 10% or 30 minutes, whichever was smaller, of the planned time, the planned time was used for determination of the pharmacokinetic parameters. In total only ten actual times were used, two for the 0.5-hour timepoint in Part 3 and eight for the 8-hour timepoint in Part 1. In Part 1, an analysis of covariance (ANCOVA) was performed to assess dose proportionality and linear kinetics of flubentylosin. In Part 2, an ANCOVA was performed to evaluate the effect of food on the bioavailability of flubentylosin. In Part 3, an ANCOVA was performed to investigate the dose proportionality of the PK variables of flubentylosin using data on Day 7 for groups with 7 days of dosing, or Day 14 for groups with 14 days of dosing. Linear mixed-effects models were used to assess the exposure-response relationship between flubentylosin concentration and $\Delta\Delta$QTcF (baseline-adjusted drug vs. placebo difference in the QT interval corrected by the Fridericia's formula).

# Results

## Disposition of subjects

In total, 78 healthy male or female subjects were exposed to flubentylosin and 22 received placebo (see Fig 1). Demographic characteristics were consistent across the three parts of the study (Table 2). A total of 88 subjects completed the study.

The 12 subjects enrolled in Part 2 (food effect) were discontinued from the study after completing Period 1 because two subjects in Period 1 experienced AEs of elevated liver enzymes that were considered possibly related to the study drug, which met a prespecified pause criterion. As a result, Period 2 of Part 2 of the study was put on hold. The flubentylosin development team hypothesized that the liver enzyme elevations were related to flubentylosin $C_{max}$ and that similar cumulative exposures to flubentylosin across 7 to 14 daily doses with a lower $C_{max}$ would be less likely to produce liver enzyme elevations. As pharmacokinetic-pharmacodynamic modeling predicted that doses of 200 to 400 mg would be efficacious, the study was amended to administer the 400-mg single dose again to a new dose group (Group 6, Part 1), to modify the MAD part to increase the dose incrementally to the target efficacious regimens, i.e., 400 mg once daily (QD) for 7 or 14 days, and to provide a longer duration of monitoring after dosing. The aim of the modification was to target predicted maximum cumulative exposure in the first MAD group that would not exceed mean exposure after a 400-mg single dose.

## PK parameters

All 78 subjects who received flubentylosin, 36 subjects in Part 1, 12 subjects in Part 2, and 30 subjects in Part 3, were included in the PK analyses.

Mean blood flubentylosin concentrations versus time profiles following administration of single doses of flubentylosin ranging from 40 to 1000 mg are shown in Fig 2, and PK parameters are presented in Table 3. Blood flubentylosin concentrations reached maximum levels at approximately 1.5 to 2 hours after dosing with a half-life less than 3 hours at doses $\leq$ 400 mg. Dose-normalized $C_{max}$ and AUC showed statistically significant ($p < 0.05$) differences

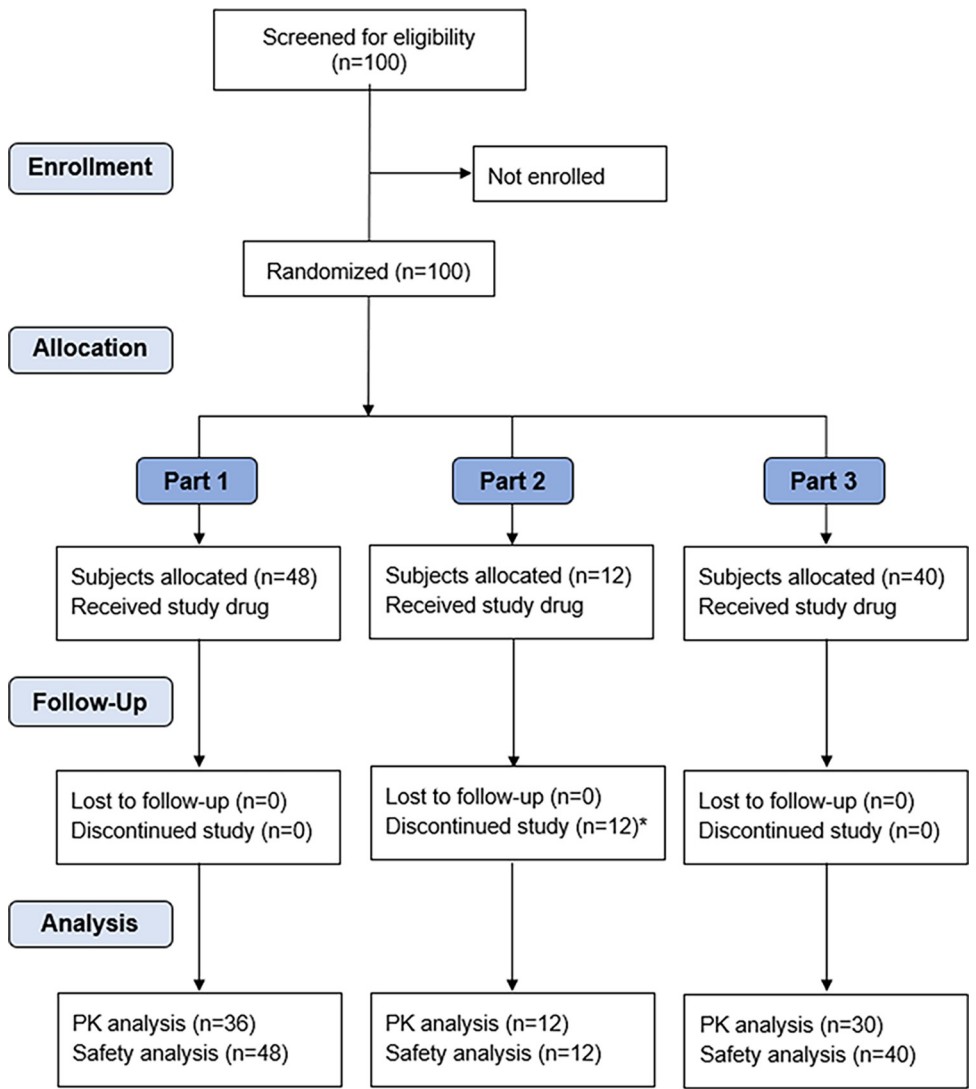

**Fig 1. Patient Disposition in Phase-I Study on Flubentylosin.**

between the highest dose (1000 mg) and the lowest dose (40 mg), and a statistically significant ($p < 0.05$) trend for a more than dose-proportional increase across the evaluated dose range from 40 to 1000 mg (Fig 3).

Mean blood flubentylosin concentrations versus time profiles after administration of flubentylosin as a single 1000-mg dose under fasting and fed conditions in Part 2 are shown in Fig 4, and PK parameters are presented in Table 4. Overall, flubentylosin exposures, based on $C_{max}$ and AUC, appeared to be similar under fasting and fed conditions (Table 5).

Mean blood flubentylosin concentrations versus time profiles following administration of multiple ascending doses of flubentylosin ranging from 100 to 400 mg QD for 7 or 14 days in Part 3 are shown in Fig 5, and PK parameters are presented in Table 6. Blood flubentylosin concentrations reached maximum levels at approximately 1.5 to 2 hours after dosing with a terminal phase elimination half-life of 2 to 3 hours across the dose groups. Renal clearance

**Table 2. Demographic Characteristics of Subjects in the Three Parts of the Phase-I Study on Flubentylosin.**

| Variable | Statistics | Part 1—SAD Study | | Part 2—Food Effect Study | Part 3—MAD Study | |
|---|---|---|---|---|---|---|
| | | Flubentylosin N = 36 | Placebo N = 12 | All subjects N = 12 | Flubentylosin N = 30 | Placebo N = 10 |
| **Sex** | Male n (%) | 22 (61.1) | 9 (75.0) | 10 (83.3) | 15 (50.0) | 5 (50.0) |
| | Female n (%) | 14 (38.9) | 3 (25.0) | 2 (16.7) | 15 (50.0) | 5 (50.0) |
| **Age (years)** | Mean (SD) | 39.5 (10.4) | 36.2 (8.33) | 36.5 (6.90) | 37.3 (9.86) | 34.8 (9.20) |
| | Range | 25–56 | 24–53 | 26–47 | 22–54 | 24–53 |
| **Ethnicity** | White n (%) | 21 (58.3) | 5 (41.7) | 4 (33.3) | 15 (50.0) | 7 (70.0) |
| | Black n (%) | 11 (30.6) | 6 (50.0) | 8 (66.7) | 12 (40.0) | 1 (10.0) |
| | Asian n (%) | 1 (2.8) | 1 (8.3) | - | 1 (3.3) | - |
| | Mixed n (%) | 3 (8.3) | - | - | 2 (6.7) | 2 (20.0) |
| **Weight (kg)** | Mean (SD) | 75.1 (9.74) | 79.0 (7.89) | 81.9 (8.90) | 72.9 (11.6) | 73.6 (11.5) |
| | Range | 59.5–99.1 | 66.1–87.4 | 62.9–95.0 | 52.8–99.4 | 50.6–90.0 |
| **Height (cm)** | Mean (SD) | 170.8 (7.73) | 174.8 (9.17) | 178.9 (5.03) | 170.9 (9.42) | 171.0 (8.64) |
| | Range | 154.4–192.6 | 160.1–185.6 | 169.4–187.9 | 153.4–187.9 | 154.8–181.3 |

MAD: multiple ascending dose; SAD: single ascending dose; SD: standard deviation

ranged from 6.3 to 8 L/h across the dose groups evaluated with less than 1% of the dose excreted unchanged in the urine.

After 7 days of once-daily dosing with flubentylosin, dose-normalized $C_{max}$ was not statistically significantly different (p = 0.1161) between the highest (400 mg) and the lowest (100 mg) dose. The dose-normalized mean $AUC_{24}$ at the 400 mg dose was statistically significantly greater than the 100 mg dose (p = 0.0050), and there was a statistically significant trend (p = 0.0025) for a more than dose proportional increase in $AUC_{24}$. The apparent terminal phase elimination rate constant (β, beta) for the 400-mg dose was statistically significantly lower (i.e. longer $t_{1/2}$) than the 100-mg dose (p = 0.0235). Additionally, the trend analysis showed a statistically significant trend for β with flubentylosin doses (p = 0.0206).

After 14 days of once-daily dosing, dose-normalized $C_{max}$, AUC, and β showed statistically significant (p < 0.05) differences in all three PK parameters between the highest (400 mg) and the lowest (200 mg) dose of flubentylosin, and a statistically significant (p < 0.05) trend in all three PK parameters for a more than dose-proportional increase in exposure was observed between the 200-mg and 400-mg doses.

## Safety

The safety analyses were performed on all 100 subjects who received flubentylosin or placebo: 36 subjects and 12 subjects, respectively, in Part 1, 12 subjects in Part 2, and 30 subjects and 10 subjects, respectively, in Part 3.

In Part 1 (SAD), 16.7% of flubentylosin-treated subjects and 0% of placebo-treated subjects experienced at least one treatment-emergent adverse event (TEAE) (Table 7). The most common (≥ 5%) TEAEs for flubentylosin-treated subjects in Part 1 were nausea (8.3%) and headache (5.6%). One subject in the 100-mg group experienced Grade 1 elevated alanine aminotransferase (ALT) on Day 3, resolving on Day 14.

In Part 2 (food effect), 58.3% of flubentylosin-treated subjects (66.7% under fasting and 50.0% under fed conditions) experienced at least one TEAE (Table 8). The most common (≥ 10%) TEAEs for flubentylosin-treated subjects in Part 2 were nausea (25.0%), diarrhea (16.7%), headache (16.7%), hepatic enzyme increased (16.7%), and vomiting (16.7%).

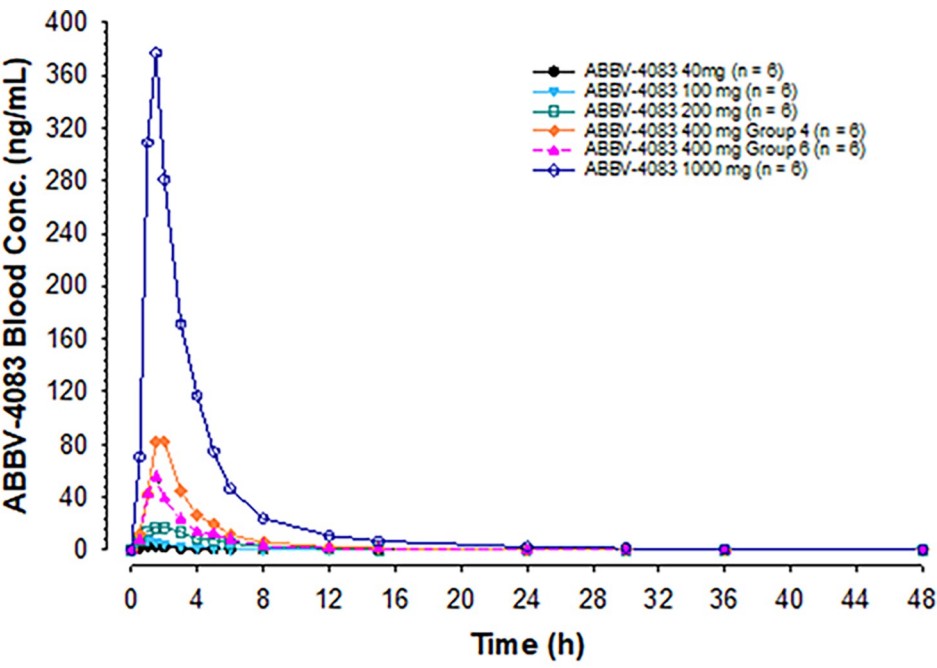

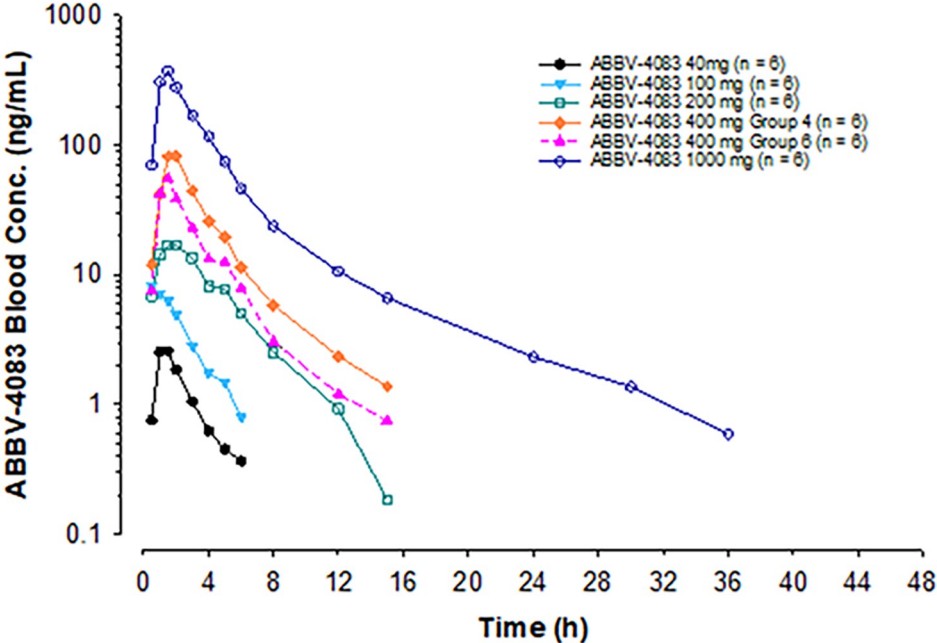

ABBV-4083 = flubentylosin

**Fig 2. Mean Blood Flubentylosin Concentrations Versus Time Profiles after Single Oral Administration of Ascending Doses from 40 mg to 1000 mg (Part 1), Linear (top) and Log-linear (bottom) Scales.**

During Period 1, Part 2 of the study, 2 subjects receiving flubentylosin under fasting conditions experienced asymptomatic and reversible Grade-1 elevated ALT and/or AST values. For those receiving flubentylosin under fed conditions, 2 subjects experienced AEs of

**Table 3. Mean PK Parameters for Flubentylosin in Part 1 (Single Ascending Doses).**

| PK Parameter (units) | Group 1 40 mg n = 6[c] | Group 2 100 mg n = 6[d] | Group 3 200 mg n = 6 | Group 4 400 mg n = 6 | Group 5 1000 mg n = 6 | Group 6 400 mg n = 6 |
|---|---|---|---|---|---|---|
| $C_{max}$ (ng/mL) | 2.57 (2.78, 50) | 8.04 (11.6, 87) | 21.6 (26.9, 83) | 80.2 (92.8, 67) | 379 (408, 43) | 50.7 (59.1, 55) |
| $T_{max}$[a] (h) | 1.5 (1.0, 3.0) | 1.5 (0.5, 3.0) | 2.0 (1.0, 3.0) | 1.8 (1.5, 2.0) | 1.5 (1.0, 2.0) | 1.5 (1.0, 2.0) |
| $T_{1/2}$[b] (h) | 2.60 (1.77) | 2.22 (0.588) | 2.40 (0.976) | 2.97 (0.604) | 6.62 (3.02) | 2.57 (0.503) |
| $AUC_t$ (ng•h/mL) | 4.46 (6.24, 96) | 16.8 (20.8, 60) | 69.7 (78.3, 56) | 241 (265, 55) | 1120 (1190, 34) | 145 (161, 46) |
| $AUC_{inf}$ (ng•h/mL) | 18.0 (18.1, 15) | 26.5 (28.5, 39) | 74.6 (83.3, 54) | 248 (272, 54) | 1130 (1210, 34) | 152 (166, 45) |
| $C_{max}$/Dose (ng/mL/mg) | 0.0642 (0.0696, 50) | 0.0804 (0.116, 87) | 0.108 (0.134, 83) | 0.200 (0.232, 67) | 0.379 (0.408, 43) | 0.127 (0.148, 55) |
| $AUC_t$/Dose (ng•h/mL/mg) | 0.112 (0.156, 96) | 0.168 (0.208, 60) | 0.348 (0.391, 56) | 0.601 (0.663, 55) | 1.12 (1.19, 34) | 0.363 (0.401, 46) |
| $AUC_{inf}$/Dose (ng•h/mL/mg) | 0.451 (0.454, 15) | 0.265 (0.285, 39) | 0.373 (0.417, 54) | 0.619 (0.680, 54) | 1.13 (1.21, 34) | 0.379 (0.416, 45) |

a. Median (range)

b. Harmonic mean (pseudo-standard deviation)

c. n = 2 for $t_{1/2}$, $AUC_{inf}$ and $AUC_{inf}$/Dose

d. n = 5 for $t_{1/2}$, $AUC_{inf}$ and $AUC_{inf}$/Dose

asymptomatic and reversible elevated liver enzymes leading to discontinuation of the study drug: one subject had Grade-2 elevations of ALT at 131 U/L (normal range 0–35 U/L) and AST at 102 U/L (normal range 9–35 U/L) six days after receiving a single 1000-mg dose of flubentylosin, and one subject had Grade-4 elevations of ALT at 425 U/L (normal range 0–35 U/L) and AST at 453 U/L (normal range 9–35 U/L) three days after receiving a single 1000-mg dose of flubentylosin. No accompanying increases in total, direct, or indirect bilirubin were observed. The investigator considered the AEs to have a reasonable possibility of being related to flubentylosin. The food effect on exposure parameters, i.e. $C_{max}$ or AUC was minimal. In view of the observed liver enzyme elevations, the single dose of 400 mg was tested again in Part 1 Group 6 with monitoring extended from 3 to 10 days. No above-normal transaminase levels were observed. The subsequent MAD portions of the study (Part 3) were conducted with the same extended monitoring period.

In Part 3 (MAD), 33.3% of flubentylosin-treated subjects and 30.0% of placebo-treated subjects experienced at least one TEAE (Table 9). The most common ($\geq$ 5%) TEAEs for flubentylosin-treated subjects in Part 3 were headache (6.7%), nausea (6.7%), palpitations (6.7%), and medical device site reaction (6.7%). Three subjects experienced asymptomatic and reversible Grade-1 elevated ALT and/or AST in the dose group receiving 100 mg QD for 7 days. None were accompanied by elevated bilirubin levels. No elevated transaminase levels were observed in the dose groups receiving 200 or 400 mg for 7 days or 200 or 400 mg QD for 14 days.

No serious adverse events related to flubentylosin were reported during the study (Table 10). One death occurred during the study: a 33-year-old female subject who died from a presumed pulmonary embolism 9 days after receiving a single 1000-mg dose of flubentylosin in Part 2 of the study. The death was deemed to be unrelated to the single dose of study drug. Although a specific etiology was not identified, it was noted that she had thrombocytosis and hypercholesterolemia at baseline and may have had a hypercoagulable state that predisposed her to thrombotic events.

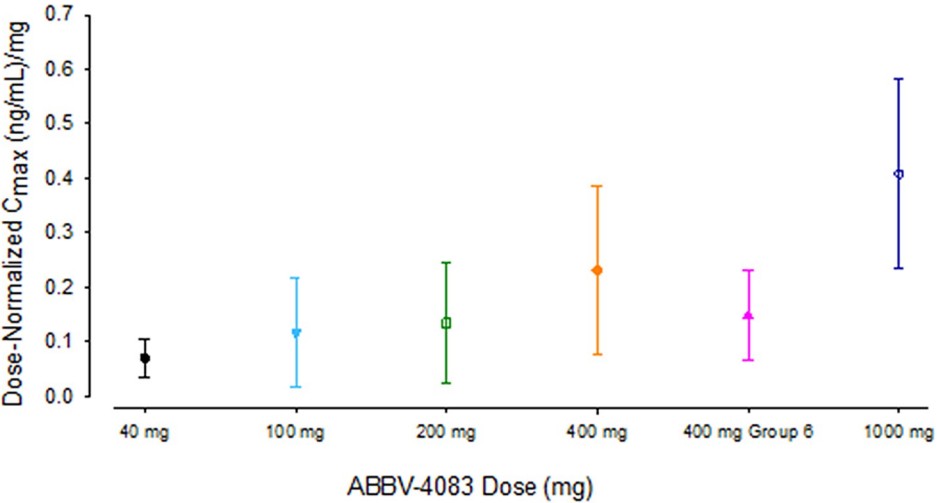

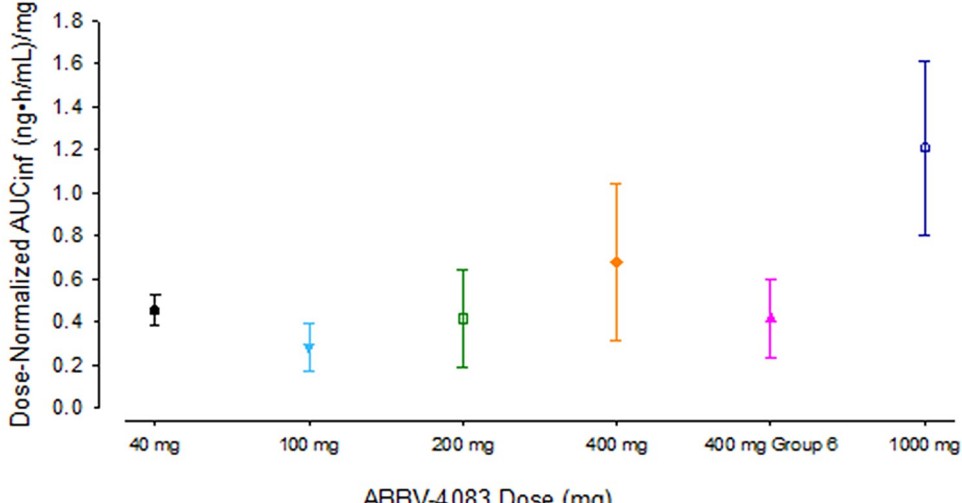

ABBV-4083 = flubentylosin

**Fig 3. Mean Dose-normalized $C_{max}$ and AUC of Flubentylosin after Single Oral Administration of Ascending Doses from 40 mg to 1000 mg (Part 1).**

Lastly, there did not appear to be a relationship between flubentylosin concentrations and changes in QTcF from baseline. The upper bounds of the 90% confidence intervals of ΔΔQTcF ranged from 2.45 to 5.01 msec, well below the 10-msec threshold, for all dose levels tested, suggesting that flubentylosin had no clinically relevant effects on QTc prolongation.

## Discussion

Via its anti-*Wolbachia* activity, the oral macrolide flubentylosin represents a promising approach to the control and treatment of onchocerciasis and LF in sub-Saharan Africa. Unlike current therapies, it is macrofilaricidal, thus acting at different stages of the parasites' life cycle. Moreover, it can be used in various geographic areas, including those where *Loa loa* is co-endemic, and does not have contraindications in specific populations such as women of child-bearing age.

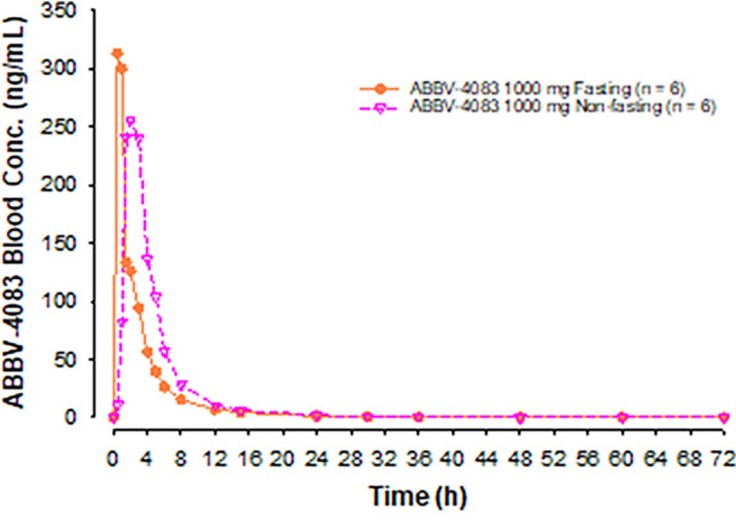

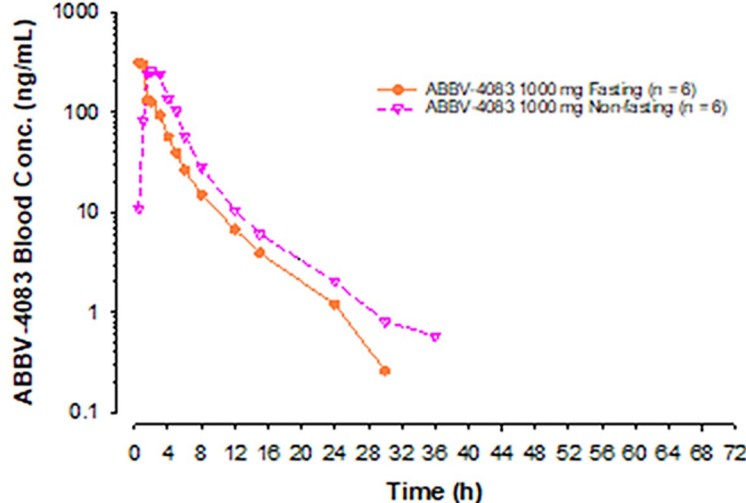

ABBV-4083 = flubentylosin

**Fig 4. Mean Blood Flubentylosin Concentrations versus Time Profiles after Administration of Flubentylosin 1000 mg under Fasting and Fed Conditions in Part 2 (Food Effect), Linear (top) and Log-linear (bottom) Scales.**

Across all three parts of the study, the PK findings showed good bioavailability of flubentylosin in human subjects, with rapid absorption and sustained exposure over its acceptably long half-life, following single- and multiple-dose administration. Exposure increased in a slightly more than dose-proportional manner across the doses. Multiple-dose administration yielded similar blood flubentylosin concentrations on Day 1 and Day 14. Flubentylosin concentrations also appeared to be similar under fasting and fed conditions, which would be a desirable attribute for a drug that will likely be used in settings where consistent dosing with relation to food cannot be assured.

Overall, the safety profile of flubentylosin in human subjects was good at doses up to and including 400 mg QD for 14 days. The majority of the reported AEs were mild to or moderate. No flubentylosin-related serious adverse events were reported. The asymptomatic and

**Table 4. Geometric Mean PK Parameters for Flubentylosin after Administration under Fasting and Fed Conditions in Part 2 (Food Effect).**

| PK Parameter (units) | Flubentylosin 1000 mg Fasting (n = 6) | Flubentylosin 1000 mg Fed (n = 6) |
|---|---|---|
| $C_{max}$ (ng/mL) | 371 (377, 18) | 320 (334, 32) |
| $T_{max}$ [a] (h) | 0.8 (0.5, 1.0) | 2.0 (1.5, 3.0) |
| $T_{\frac{1}{2}}$ [b] (h) | 4.41 (1.34) | 5.50 (2.85) |
| $AUC_t$ (ng•h/mL) | 752 (794, 42) | 1040 (1100, 39) |
| $AUC_{inf}$ (ng•h/mL) | 764 (805, 42) | 1050 (1110, 38) |

a. Median (range)

b. Harmonic mean (pseudo-standard deviation)

**Table 5. Relative Bioavailability of Flubentylosin after Administration under Fasting and Fed Conditions in Part 2 (Food Effect).**

| Regimen | PK Parameter (units-) | Central Value[a] | | Relative Bioavailability | |
|---|---|---|---|---|---|
| | | Test | Reference | Point Estimate[b] | 90% CI[c] |
| Fasting | $C_{max}$ (ng/mL) | 320 | 371 | 0.861 | 0.649, 1.141 |
| Fed | $AUC_t$ (ng•h/mL) | 974 | 801 | 1.215 | 0.848, 1.743 |
| | $AUC_{inf}$ (ng•h/mL) | 987 | 813 | 1.214 | 0.850, 1.734 |

a. Exponentiation of the least squares means for logarithms

b. Exponentiation of the difference (test minus reference) of the least squares means for logarithms

c. Exponentiation of the endpoints of confidence intervals for the difference of the least squares means for logarithms

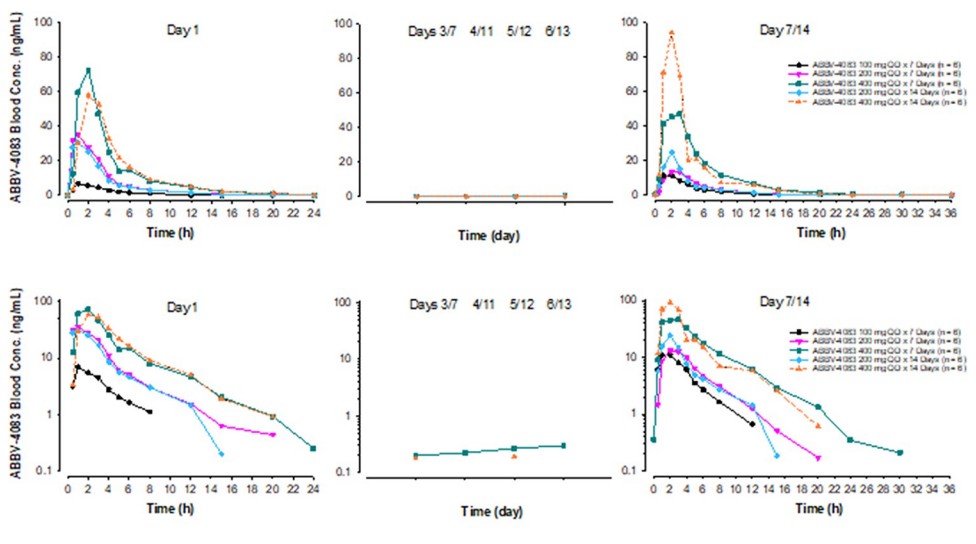

ABBV-4083 = flubentylosin

**Fig 5. Mean Blood Flubentylosin Concentrations versus Time Profiles after Administration of Multiple Ascending Doses of Flubentylosin in Part 3 (Upper Panels Linear Scale—Lower Panels Log-Linear Scale)**

**Table 6. Geometric Mean (Arithmetic Mean, % CV) PK Parameters for Flubentylosin after Administration of Multiple Ascending Doses in Part 3.**

| PK Parameter (units) | Flubentylosin Multiple Ascending Dose—Day 7 or Day 14 | | | | |
|---|---|---|---|---|---|
| | Group 8 100 mg QD 7 Days (N = 6) | Group 9 200 mg QD 7 Days (N = 6) | Group 10 200 mg QD 14 Days (N = 6) | Group 11 400 mg QD 7 Days (N = 6) | Group 12 400 mg QD 14 Days (N = 6) |
| $C_{max}$ (ng/mL) | 10.6 (13.9, 73) | 12.3 (16.5, 91) | 22.6 (24.7, 46) | 62.7 (68.3, 40) | 97.1 (101, 32) |
| $T_{max}$ [a] (h) | 1.5 (1.0, 3.0) | 1.8 (1.5, 3.0) | 1.5 (1.0, 1.5) | 1.5 (1.0, 3.0) | 1.5 (1.0, 1.5) |
| $T_{\frac{1}{2}}$ [b] (h) | 2.11 (0.498) | 2.38 (0.906) | 2.38 (0.355) | 3.41 (1.19) | 3.30 (0.761) |
| $AUC_{24}$ (ng•h/mL) | 31.0 (37.2, 59) | 39.6 (53.7, 91) | 54.9 (60.3, 44) | 191 (215, 63) | 232 (237, 23) |
| CL/F (L/h) | 3230 (3980, 74) | 5050 (6400, 63) | 3640 (4040, 52) | 2090 (2270, 38) | 1720 (1750, 21) |
| $C_{max}$/Dose (ng/mL/mg) | 0.106 (0.139, 73) | 0.0617 (0.0823, 91) | 0.113 (0.124, 46) | 0.157 (0.171, 40) | 0.243 (0.253, 32) |
| $C_{max}$ $R_{ac}$ [a] | 1.83 (1.27, 4.16) | 0.595 (0.253, 0.777) | 0.941 (0.197, 1.26) | 0.879 (0.743, 1.43) | 1.61 (1.11, 2.07) |
| $AUC_{24}$/Dose (ng•h/mL/mg) | 0.310 (0.372, 59) | 0.198 (0.269, 91) | 0.275 (0.301, 44) | 0.478 (0.539, 63) | 0.581 (0.593, 23) |
| $AUC_{24}$ $R_{ac}$ [a] | 2.03 (1.40, 5.20) | 0.632 (0.427, 1.15) | 0.954 (0.446, 1.11) | 1.07 (0.888, 1.28) | 1.24 (0.988, 1.69) |
| CLr (L/h) | 6.28 (6.46, 27) | 7.01 (7.38, 33) | 7.23 (7.34, 18) | 6.75 (7.06, 32) | 8.01 (8.09, 15) |
| $F_e$ (%) | 0.195 (0.213, 44) | 0.139 (0.172, 68) | 0.199 (0.214, 40) | 0.323 (0.342, 36) | 0.466 (0.476, 24) |

a. Median (range)

b. Harmonic mean (pseudo-standard deviation)

Note: $R_{ac}$ calculated as the ratio of $C_{max}$ or $AUC_{24}$ from Day 7 or Day 14 to Day 1

QD: once daily

**Table 7. Treatment-emergent Adverse Events Reported with Flubentylosin in Part 1 (Single Ascending Doses), presented by System-organ Class.**

| System Organ Class | Placebo N = 12 n (%) | Flubentylosin | | | | | | Total N = 36 n (%) |
|---|---|---|---|---|---|---|---|---|
| | | Group 1 40 mg N = 6 n (%) | Group 2 100 mg N = 6 n (%) | Group 3 200 mg N = 6 n (%) | Group 4 400 mg N = 6 n (%) | Group 5 1000 mg N = 6 n (%) | Group 6 400 mg N = 6 n (%) | |
| *Any TEAE* | *0* | *1 (16.7)* | *1 (16.7)* | *1 (16.7)* | *0* | *2 (33.3)* | *1 (16.7)* | *6 (16.7)* |
| Gastrointestinal disorders | 0 | 1 (16.7) | 0 | 0 | 0 | 2 (33.3) | 1 (16.7) | 4 (11.1) |
| Constipation | 0 | 0 | 0 | 0 | 0 | 0 | 1 (16.7) | 1 (2.8) |
| Nausea | 0 | 1 (16.7) | 0 | 0 | 0 | 2 (33.3) | 0 | 3 (8.3) |
| Vomiting | 0 | 0 | 0 | 0 | 0 | 1 (16.7) | 0 | 1 (2.8) |
| Nervous system disorders | 0 | 1 (16.7) | 1 (16.7) | 1 (16.7) | 0 | 0 | 0 | 3 (8.3) |
| Dizziness | 0 | 0 | 1 (16.7) | 0 | 0 | 0 | 0 | 1 (2.8) |
| Headache | 0 | 1 (16.7) | 0 | 1 (16.7) | 0 | 0 | 0 | 2 (5.6) |
| Somnolence | 0 | 0 | 1 (16.7) | 0 | 0 | 0 | 0 | 1 (2.8) |

Subjects are counted only once in each row, regardless of how many events they experienced.

**Table 8. Treatment-emergent Adverse Events Reported with Flubentylosin in Part 2 (Food Effect), presented by System-organ Class.**

| System Organ Class | Flubentylosin | | |
|---|---|---|---|
| | Fasting 1000 mg N = 6 n (%) | Fed 1000 mg N = 6 n (%) | Total N = 12 n (%) |
| *Any TEAE* | *4 (66.7)* | *3 (50.0)* | *7 (58.3)* |
| Gastrointestinal disorders | 3 (50.0) | 0 | 3 (25.0) |
| Diarrhoea | 2 (33.3) | 0 | 2 (16.7) |
| Nausea | 3 (50.0) | 0 | 3 (25.0) |
| Vomiting | 2 (33.3) | 0 | 2 (16.7) |
| Investigations | 0 | 2 (33.3) | 2 (16.7) |
| Hepatic enzyme increased | 0 | 2 (33.3) | 2 (16.7) |
| Musculoskeletal and connective tissue disorders | 1 (16.7) | 0 | 1 (8.3) |
| Neck pain | 1 (16.7) | 0 | 1 (8.3) |
| Nervous system disorders | 2 (33.3) | 0 | 2 (16.7) |
| Headache | 2 (33.3) | 0 | 2 (16.7) |
| Respiratory, thoracic and mediastinal disorders | 0 | 1 (16.7) | 1 (8.3) |
| Pulmonary embolism | 0 | 1 (16.7) | 1 (8.3) |

Subjects are counted only once in each row, regardless of how many events they experienced.

reversible increases in ALT and/or AST seen after a single 1000-mg dose of flubentylosin were not present in subjects given multiple doses of flubentylosin up to 400 mg QD for 14 days. None of the subjects with ALT and/or AST elevations had concurrent bilirubin elevations, which indicate severe drug-induced liver injury, and none developed severe and/or life-threatening outcomes such as liver failure.

Flubentylosin is a macrolide antibacterial, a widely used therapeutic class that has been found to be associated with acute liver injury, initially in spontaneous reports and case series, and more recently in a systematic review and meta-analysis [18]. The mechanism underlying liver injury with macrolides has not yet been fully elucidated, however oxidative stress, mitochondrial toxicity, and bile acid accumulation have been put forward as possible explanations [19]. In this study, 2 subjects experienced AEs of elevated liver enzymes leading to discontinuation of the study drug, both after receiving a single 1000-mg dose of flubentylosin. The development team hypothesized that the AEs were $C_{max}$-related, and the study was amended, leading to deletion of the single-dose 800-mg and 1200-mg groups initially planned in the protocol. The 400-mg dose group was repeated, the MAD part was modified, and no other AEs of elevated liver enzymes were observed in doses up to 400 mg QD for 14 days with 10 days of post-dose monitoring.

Thus, this three-part, first-in-human Phase-I study provided valuable information on the safety, PK profile, dose proportionality and linearity of flubentylosin in healthy human subjects, as well as the food effect. It also enabled us to identify a dose and dosing regimens, i.e. 400 mg of flubentylosin for 7 or 14 days, that are expected to achieve efficacious exposures and provide an adequate safety margin. If safety and antifilarial activity are confirmed in clinical trials, an effective therapy for onchocerciasis and/or lymphatic filariasis that requires no more than 14 days of dosing would be a substantial improvement on currently available treatments. Phase-II studies using these regimens in patients with onchocerciasis are currently ongoing.

**Table 9. Treatment-emergent Adverse Events Reported with Flubentylosin in Part 3 (Multiple Ascending Doses), presented by System-organ Class.**

| System Organ Class | Placebo N = 10 n (%) | Flubentylosin | | | | | Total N = 30 n (%) |
|---|---|---|---|---|---|---|---|
| | | Group 8 100 mg ´ 7 d N = 6 N (%) | Group 9 200 mg ´ 7 d N = 6 N (%) | Group 10 200 mg ´ 14 d N = 6 N (%) | Group 11 400 mg ´ 7 d N = 6 N (%) | Group 12 400 mg ´ 14 d N = 6 N (%) | |
| *Any TEAE* | *3 (30.0)* | *4 (66.7)* | *1 (16.7)* | *1 (16.7)* | *1 (16.7)* | *3 (50.0)* | *10 (33.3)* |
| Cardiac disorders | 0 | 0 | 0 | 0 | 0 | 2 (33.3) | 2 (6.7) |
| Palpitations | 0 | 0 | 0 | 0 | 0 | 2 (33.3) | 2 (6.7) |
| Eye disorders | 0 | 0 | 0 | 1 (16.7) | 0 | 0 | 1 (3.3) |
| Dry eye | 0 | 0 | 0 | 1 (16.7) | 0 | 0 | 1 (3.3) |
| Gastrointestinal disorders | 1 (10.0) | 1 (16.7) | 0 | 0 | 0 | 2 (33.3) | 3 (10.0) |
| Abdominal pain | 0 | 1 (16.7) | 0 | 0 | 0 | 0 | 1 (3.3) |
| Constipation | 0 | 0 | 0 | 0 | 0 | 1 (16.7) | 1 (3.3) |
| Nausea | 1 (10.0) | 0 | 0 | 0 | 0 | 2 (33.3) | 2 (6.7) |
| Vomiting | 1 (10.0) | 0 | 0 | 0 | 0 | 0 | 0 |
| General disorders and administration site disorders | 0 | 0 | 1 (16.7) | 0 | 1 (16.7) | 0 | 2 (6.7) |
| Medical device site reaction | 0 | 0 | 1 (16.7) | 0 | 1 (16.7) | 0 | 2 (6.7) |
| Infections & infestations | 0 | 2 (33.3) | 0 | 0 | 0 | 0 | 2 (6.7) |
| Body tinea | 0 | 1 (16.7) | 0 | 0 | 0 | 0 | 1 (3.3) |
| Gastroenteritis | 0 | 1 (16.7) | 0 | 0 | 0 | 0 | 1 (3.3) |
| Injury, poisoning and procedural complications | 1 (10.0) | 0 | 0 | 0 | 0 | 0 | 0 |
| Contusion | 1 (10.0) | 0 | 0 | 0 | 0 | 0 | 0 |
| Musculoskeletal and connective tissue disorders | 1 (10.0) | 0 | 0 | 0 | 0 | 1 (16.7) | 1 (3.3) |
| Musculoskeletal chest pain | 1 (10.0) | 0 | 0 | 0 | 0 | 0 | 0 |
| Musculoskeletal pain | 0 | 0 | 0 | 0 | 0 | 1 (16.7) | 1 (3.3) |
| Nervous system disorders | 0 | 1 (16.7) | 0 | 0 | 0 | 1 (16.7) | 2 (6.7) |
| Headache | 0 | 1 (16.7) | 0 | 0 | 0 | 1 (16.7) | 2 (6.7) |
| Psychiatric disorders | 1 (10.0) | 0 | 0 | 0 | 0 | 0 | 0 |
| Acute stress disorder | 1 (10.0) | 0 | 0 | 0 | 0 | 0 | 0 |
| Reproductive system and breast disorders | 0 | 1 (16.7) | 0 | 0 | 0 | 1 (16.7) | 2 (6.7) |
| Dysmenorrhoea | 0 | 1 (16.7) | 0 | 0 | 0 | 0 | 1 (3.3) |
| Vaginal discharge | 0 | 0 | 0 | 0 | 0 | 1 (16.7) | 1 (3.3) |
| Respiratory, thoracic and mediastinal disorders | 1 (10.0) | 0 | 0 | 0 | 0 | 1 (16.7) | 1 (3.3) |
| Dyspnoea exertional | 0 | 0 | 0 | 0 | 0 | 1 (16.7) | 1 (3.3) |
| Nasal congestion | 1 (10.0) | 0 | 0 | 0 | 0 | 0 | 0 |

d: days

Subjects are counted only once in each row, regardless of how many events they experienced.

**Table 10. Adverse Events Reported in the Phase-I Study.**

| Adverse Event, n (%) | Single Ascending Dose | | Food Effect | Multiple Ascending Dose | | Flubentylosin |
|---|---|---|---|---|---|---|
| | Flubentylosin (N = 36) | Placebo (N = 12) | Flubentylosin (N = 12) | Flubentylosin (N = 30) | Placebo (N = 10) | Total (N = 78) |
| Any AE | 6 (16.7) | 0 (0) | 7 (58.3) | 10 (33.3) | 3 (30) | 23 (29.5) |
| Flubentylosin-related* AE | 4 (11.1) | - | 5 (41.7)† | 2 (6.7) | - | 11 (14.1) |
| Serious AE | 0 | 0 | 1 (8.3)†† | 0 | 0 | 1 (1.3)†† |
| Flubentylosin-related* Serious AE | 0 | - | 0 | 0 | - | 0 |
| Death | 0 | 0 | 1 (8.3)†† | 0 | 0 | 1 (1.3) †† |

AE: adverse event

* Relatedness of AEs to flubentylosin as determined by study investigator

† Two subjects with AE of elevated serum liver enzymes

†† Unconfirmed pulmonary embolus 8 days after administration of study drug, deemed not related to study drug.

## Supporting information

**S1 Data. Contains all pharmacokinetic data for the Phase 1 study.**
(XLS)

## Acknowledgments

We thank the subjects who participated in the study, as well as the invstigator and clinical research staff from the study center. Medical writing support was provided by Graham Smith, and study support from Dr. Sabine Specht from Drugs for Neglected Diseases initiative.

## Author Contributions

**Conceptualization:** Negar Alami, David C. Carter, Kennan C. Marsh, Dale J. Kempf.

**Data curation:** Negar Alami, Nisha V. Kwatra, Weihan Zhao, Linda Snodgrass, Ariel R. Porcalla, Cheri E. Klein, Daniel E. Cohen.

**Formal analysis:** Nisha V. Kwatra, Weihan Zhao, Cheri E. Klein.

**Investigation:** Negar Alami, David C. Carter, Loretta Gallenberg, Melina Neenan, Robert A. Carr.

**Methodology:** Negar Alami, Nisha V. Kwatra, Cheri E. Klein, Robert A. Carr, Kennan C. Marsh, Dale J. Kempf.

**Project administration:** Melina Neenan.

**Resources:** Loretta Gallenberg, Robert A. Carr, Kennan C. Marsh, Dale J. Kempf.

**Software:** Weihan Zhao.

**Supervision:** Negar Alami, Daniel E. Cohen, Kennan C. Marsh, Dale J. Kempf.

**Validation:** Melina Neenan.

**Visualization:** Linda Snodgrass, Dale J. Kempf.

**Writing – original draft:** Negar Alami.

**Writing – review & editing:** Daniel E. Cohen, Dale J. Kempf.

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
