## [Decision Letter · Decision Letter 0]

5 Apr 2022

Dear Dr. Alami,

Thank you very much for submitting your manuscript "A Phase-I pharmacokinetic, safety and food effect study on flubentylosin, a novel analog of Tylosin-A having potent anti-Wolbachia activity and antifilarial activity" for consideration at PLOS Neglected Tropical Diseases. As with all papers reviewed by the journal, your manuscript was reviewed by members of the editorial board and by several independent reviewers. In light of the reviews (below this email), we would like to invite the resubmission of a significantly-revised version that takes into account the reviewers' comments. 

Authors should consider the comments raised by both reviewers adding more details about Wolbachia in filarioids in the introduction and reorganization of the discussion section (see reviewer 2)

We cannot make any decision about publication until we have seen the revised manuscript and your response to the reviewers' comments. Your revised manuscript is also likely to be sent to reviewers for further evaluation.

Sincerely,

Domenico Otranto

Deputy Editor

Domenico Otranto

Deputy Editor

Authors should consider the comments raised by both reviewers adding more details about Wolbachia in filarioids in the introduction and reorganization of the discussion section (see reviewer 2)

Reviewer's Responses to Questions

**Key Review Criteria Required for Acceptance?**

**Methods**

-Are the objectives of the study clearly articulated with a clear testable hypothesis stated?

-Is the study design appropriate to address the stated objectives?

-Is the population clearly described and appropriate for the hypothesis being tested?

-Is the sample size sufficient to ensure adequate power to address the hypothesis being tested?

-Were correct statistical analysis used to support conclusions?

-Are there concerns about ethical or regulatory requirements being met?

Reviewer #1: The manuscript describes the results of a phase-1 safety trial with flubentylosin, which targets Wolbachia in filarial nematodes.

General comments. The manuscript is well written; the premise is correct (the need to evaluate safety profile of a potentially important drug for the control of filarial disease); the Methods are described clearly and are adequate;

Reviewer #2: (No Response)

**Results**

-Does the analysis presented match the analysis plan?

-Are the results clearly and completely presented?

-Are the figures (Tables, Images) of sufficient quality for clarity?

Reviewer #1: the Results are presented with excellent graphical support; the Discussion could put a bit more emphasis on the previous studies cited (the development of the drug/modifications to increase bioavailability following oral administration; results obtained on Wolbachia depletion in the murine model). Also a nice way to remind readers of the importance of the past A-WOL project).

This Reviewer recommends publication following very minor revision.

Abstract. Seventy-eight healthy subjects: replace with “healthy adults”.

Reviewer #2: (No Response)

**Conclusions**

-Are the conclusions supported by the data presented?

-Are the limitations of analysis clearly described?

-Do the authors discuss how these data can be helpful to advance our understanding of the topic under study?

-Is public health relevance addressed?

Reviewer #1: the Discussion could put a bit more emphasis on the previous studies cited (the development of the drug/modifications to increase bioavailability following oral administration; results obtained on Wolbachia depletion in the murine model). Also a nice way to remind readers of the importance of the past A-WOL project).

Reviewer #2: (No Response)

**Editorial and Data Presentation Modifications?**

Reviewer #1: This Reviewer recommends publication following very minor revision.

Abstract. Seventy-eight healthy subjects: replace with “healthy adults”.

Reviewer #2: (No Response)

**Summary and General Comments**

Reviewer #1: The manuscript describes the results of a phase-1 safety trial with flubentylosin, which targets Wolbachia in filarial nematodes.

General comments. The manuscript is well written; the premise is correct (the need to evaluate safety profile of a potentially important drug for the control of filarial disease); the Methods are described clearly and are adequate; the Results are presented with excellent graphical support; the Discussion could put a bit more emphasis on the previous studies cited (the development of the drug/modifications to increase bioavailability following oral administration; results obtained on Wolbachia depletion in the murine model). Also a nice way to remind readers of the importance of the past A-WOL project).

T

Reviewer #2: The manuscript PNTD-D-22-00302, entitled “A Phase-I pharmacokinetic, safety and food effect 1 study on flubentylosin, a novel analog 2 of Tylosin-A having potent anti-Wolbachia activity and antifilarial activity” is a Phase-I pharmacokinetic, safety and food-effect study on single and multiple ascending doses of the macrolide antibacterial drug, flubentylosin (ABBV-4083), targeting filarial endosymbiont, Wolbachia for the effective control of filariasis and onchocerciasis. 

Here in this article, authors are assessing the safety and pharmacokinetics of flubentylosin in healthy subjects, to identify a dose and dosing regimen for the future studies. In general, the introduction section should include the role of Wolbachia in filarids and its imunopathological effects in filariasis. Here you are targeting Wolbachia for the treatment and hence more importance should be given for this endosymbiont. Discussion has to be rewritten. Avoid the repetition of results in discussion and discuss your results in detail. References has to be uniformly formatted. Please follow the authors guidelines of the journal while formatting. Please recheck the text carefully for grammatical and formatting mistakes. Please see the attached PDF of the manuscript for my detailed comments for the authors.

PLOS authors have the option to publish the peer review history of their article (what does this mean?). If published, this will include your full peer review and any attached files.

Reviewer #1: Yes: LAURA HELEN KRAMER

Reviewer #2: Yes: Ranju Manoj
---

## [Decision Letter · Decision Letter 1]

19 May 2023

Dear Alami,

We are pleased to inform you that your manuscript 'A Phase-I pharmacokinetic, safety and food-effect study on flubentylosin, a novel analog of Tylosin-A having potent anti-Wolbachia and antifilarial activity' has been provisionally accepted for publication in PLOS Neglected Tropical Diseases.

Best regards,

Cinzia Cantacessi

Section Editor

Domenico Otranto

Academic Editor

Reviewer's Responses to Questions

**Key Review Criteria Required for Acceptance?**

**Methods**

-Are the objectives of the study clearly articulated with a clear testable hypothesis stated?

-Is the study design appropriate to address the stated objectives?

-Is the population clearly described and appropriate for the hypothesis being tested?

-Is the sample size sufficient to ensure adequate power to address the hypothesis being tested?

-Were correct statistical analysis used to support conclusions?

-Are there concerns about ethical or regulatory requirements being met?

Reviewer #2: The objectives of the study was clearly articulated with testable hypothesis.

The design was appropriate and have addressed all the objectives

The sample size was adequate and the statistical analysis was appropriate

All ethical requirements were met in the study.

**Results**

-Does the analysis presented match the analysis plan?

-Are the results clearly and completely presented?

-Are the figures (Tables, Images) of sufficient quality for clarity?

Reviewer #2: The results presented matched with the plan of study

Authors were successful in presenting the results, clearly and concisely

The quality of figures and tables were good

**Conclusions**

-Are the conclusions supported by the data presented?

-Are the limitations of analysis clearly described?

-Do the authors discuss how these data can be helpful to advance our understanding of the topic under study?

-Is public health relevance addressed?

Reviewer #2: Conclusions were supported by the presented results and the limitations were clearly mentioned. Authors have discussed the significance of the data and its public health significance.

**Editorial and Data Presentation Modifications?**

Reviewer #2: Accept the article with minor changes

**Summary and General Comments**

Reviewer #2: I am pleased to inform that the R1 has considerably improved the paper. Authors have carefully addressed the reviewer’s comments. Hence in my opinion this paper can be accepted for publication in PLOS Neglected Tropical Diseases. Please see the pdf for some minor corrections.

PLOS authors have the option to publish the peer review history of their article (what does this mean?). If published, this will include your full peer review and any attached files.

Reviewer #2: No

---

## [Editor Report · Acceptance letter]

30 Jun 2023

Dear Alami,

We are delighted to inform you that your manuscript, "A Phase-I pharmacokinetic, safety and food-effect study on flubentylosin, a novel analog of Tylosin-A having potent anti-Wolbachia and antifilarial activity," has been formally accepted for publication in PLOS Neglected Tropical Diseases.

Best regards,

Shaden Kamhawi

co-Editor-in-Chief

Paul Brindley

co-Editor-in-Chief
